

# Effects of soil pH on the growth, soil nutrient composition, and rhizosphere microbiome of *Ageratina adenophora*

Yun Xia[1,2], Junna Feng[2], Hongbo Zhang[2], Deyu Xiong[2], Lingdong Kong[2], Robert Seviour[3] and Yunhong Kong[4]

[1] Yunnan Urban Agricultural Engineering & Technological Research Centre, Kunming University, Kunming, Yunnan Province, China
[2] School of Agriculture and Biotechnology, Kunming University, Kunming, Yunnan, China
[3] Microbiology Department, La Trobe University, Melbourne, Vic, Australia
[4] Kunming Key Laboratory of Hydro-ecology Restoration of Dianchi Lake, Kunming University, Kunming, Yunnan, China

Corresponding authors
Yun Xia, xiayun22@hotmail.com
Yunhong Kong, kongyunhong@hotmail.com

## ABSTRACT

*Ageratina adenophora* is an invasive weed species found in many countries. Methods to control the spread of this weed have been largely unsuccessful. Soil pH is the most important soil factor affecting the availability of nutrients for plant and impacting its growth. Understanding the mechanisms of the influence of soil pH on the growth of *A. adenophora* may help to develop effective control measures. In this study, we artificially changed the soil pH in pot experiments for *A. adenophora*. We studied the effects of acidic (pH 5.5), weakly acidic (pH 6.5), neutral (pH 7.2), and alkaline (pH 9.0) soils on the growth, availability of soil nutrients, activity of antioxidant enzymes, levels of redox markers in the leaves, and the structure and diversity of the rhizosphere microbiome. Soil with a pH 7.2 had a higher (47.8%) below-ground height versus soils of pH 5.5 at day 10; plant had a higher (11.3%) above-ground height in pH 7.2 soils than pH 9.0 soils at day 90; no differences in the fresh and dry weights of its above- and belowground parts, plant heights, and root lengths were observed in plants growing in acid, alkaline, or neutral pH soil were observed at day 180. Correspondingly, the antioxidant enzymes SOD (superoxide dismutase), POD (peroxidase), CAT (catalase) and redox markers GSH (glutathione) and MDA (malondialdehyde) were measured in the leaves. Significant differences existed in the activities of CAT and the levels of GSH between those growing in acidic and alkaline soils and those in neutral pH soil at day 90; however, only lower (36.8%) CAT activities in those grown at pH 5.5 than those grown at pH 7.2 were found at day 180. Similarly, significant differences in available P (16.89 *vs* 3.04 mg Kg$^{-1}$) and total K (3.67 *vs* 0.96 mg Kg$^{-1}$), total P (0.37 *vs* 0.25 g Kg$^{-1}$) and total N (0.45 *vs* 1.09 g Kg$^{-1}$) concentrations were found between the rhizosphere soils of *A. adenophora* grown at pH 9.0 and 7.2 at day 90; no such differences were seen at day 180. High throughput analyses of the 16S rRNA and ITS fragments showed that the rhizosphere microbiome diversity and composition under different soil pH conditions changed over 180 days. The rhizosphere microbiomes differed in diversity, phylum, and generic composition and population interactions under acid and alkaline conditions versus those grown in neutral soils. Soil pH had a greater impact on the diversity and composition of the prokaryotic rhizosphere communities than those of the fungal communities. *A. adenophora* responded successfully to pH stress by

changing the diversity and composition of the rhizosphere microbiome to maintain a balanced nutrient supply to support its normal growth. The unusual pH tolerance of *A. adenophora* may be one crucial reason for its successful invasion. Our results suggest that attempts use soil pH to control its invasion by changing the soil pH (for example, using lime) will fail.

## INTRODUCTION

*Ageratina adenophora* (Spreng.) is a perennial semi-shrubby herb native to Mexico and Costa Rica. It is one of the major invasive weed species in Africa, Oceania, and Asia (*Zhang et al., 2022*). It sexually reproduces by producing many seeds and has a strong ability to asexually reproduce through the roots and stems (*Li et al., 2007*). It is widely distributed throughout southwest China after its invasion of the Yunnan Province in China from the border of Myanmar in the 1940s. It is found vertically between altitudes of 165–2,915 m and at tropical, subtropical, central and northern subtropical, warm temperate and temperate zones. It grows in open areas, forest margins, riverbanks, roadsides, grasslands, crop fields, pastures, woodlands, limestone shrubs, plantations, arid wastelands, and surrounding agricultural areas with different soil types (*Wang et al., 2011*). Its bud bursting stage typically begins in late November and the first blossom happens in mid- to late-February of the following year. Its growth stage starts from May to September with the fastest growth period from July to August. Flower-bud differentiation occurs in November (*Ping, Sang & Ma, 2005*). *A. adenophora* forms a single-dominant community in an invaded area, thus reducing the biodiversity and destroying the balance of the ecosystems (*McGeoch et al., 2010*). This weed has led to an annual loss of 150 million US dollars from livestock production decline and 400 million US dollars from the service function of grassland ecosystems in China (*Xu et al., 2006*). To date, the effectiveness of various control methods including artificial mechanical and chemical control, introduction of natural enemy controls, and biological substitution control have been largely unsuccessful (*Poudel et al., 2019*).

The influence of soil pH on the growth of invasive plants has attracted considerable attention. The soil pH plays an important role in the process of plant growth and development (*Alam, Naqvi & Ansari, 1999*; *Brady & Weil, 1999*). It affects several important soil biological and physicochemical processes including the mineralization of soil organic matter, microbial enzyme activities, ammonia volatilization, bacterial nitrification, and denitrification. All these processes are related to the survival and migration of nutrients in the soil, and thus their availability to plants (reviewed by *Neina, 2019*). Nitrogen (N) is an important plant nutrient, and is most readily available to plants where soil pH is higher than 5.5. In acidic soils, nitrification is inhibited, thus reducing the availability of nitrate. Under these conditions, plants must use ammonia as their source of N, thereby reducing N utilization efficiency (*Zebarth et al., 2015*). Maximum phosphorus availability occurs when

the soil pH ranges between 6–7. In acidic soils aluminum and iron, which form strong bonds with phosphate, are present, while at higher pH when calcium is the dominant cation, soil phosphate tends to convert to insoluble calcium phosphate (*Devau et al., 2009*). Available potassium (K) decreases with any increase in soil pH, (*Liu et al., 2020*). The ideal soil pH for plant growth is between 6.5 and 7.5. Soils that are too acidic or alkaline can negatively affect the physical properties of the soil and reduce the availability of nutrients to plants (*Brady & Weil, 1999*). Many studies have demonstrated that the application of lime to acidic soils neutralizes excessive hydrogen ions and raises soil pH, which results in greater crop productivity (*Zhang et al., 2023*). Understanding the mechanism of the influence of soil pH on plant growth is of theoretical and practical importance for the amelioration of soils with acid–base imbalances. It may also lead to improvements of soil fertility, better crop production, and the prevention and control of invasive plants (*Soti et al., 2015*).

The soil microbiome is responsible for the decomposition and transformation of soil nutrients, which in turn affect their uptake and utilization by the plant (*Neina, 2019*). Changes in soil pH can affect its biomass levels, diversity and structure (*Feng et al., 2023*; *Guo et al., 2022*; *Mod et al., 2021*; *Schlatter et al., 2020*; *Siles & Margesin, 2016*). Fungi dominate in low pH soils, while high pH soils favour the bacteria (*Alexander, 1977*). The ratio of fungi:bacteria in soil decreases with an increase in soil pH. At pH 3, this ratio is about 9, but at pH 7, falls to about 2, and soil microbial activity is inhibited at a pH less than 4.5 (*Rousk, Brookes & Bååth, 2009*). High-throughput DNA sequencing technology reveals that temperature, geographical location and other factors may affect the composition of the soil microbiome, however, the soil pH is the most important parameter (*Chen et al., 2020*; *Feng et al., 2023*; *Fierer & Jackson, 2006*; *Siles & Margesin, 2016*). *A. adenophora* has strong allelopathy and competitiveness and changes the diversity and composition of the microbiome in the invaded soil (*Kong et al., 2017*; *Xia et al., 2021*; *Xiao, Schaefer & Yang, 2017*). This invasive weed improves the composition of soil nutrient elements (*Zhao et al., 2019*) making it beneficial to support its own growth, while inhibiting or reducing the growth and competitiveness of adjacent native plants (*Wan et al., 2010*). However, it is still unclear how soil pH affects its rhizosphere microbial community diversity and composition.

In this study, pot experiments were performed to examine what effects soil pH might have on the growth of *A. adenophora* and how it affects availability of soil nutrients, antioxidant enzyme activities of its leaves, and the diversity, composition, and interactions of its rhizosphere microbiome. Such data will help to develop effective control measures for *A. adenophora* growth and its ecological impact.

## MATERIALS & METHODS

### Preparation of soils with different pH values for pot experiments

Original soil was collected from a scenery orchard located at Kunming University, Kunming, China (24°58′53″N, 102°47′54″E), which had not been invaded by *A. adenophora*. Soil samples were sieved to two mm mesh to remove plant roots and debris, and thoroughly

homogenized and air dried. To ensure sufficient nutrients for *A. adenophora* growth, the soil sample was mixed with 1:1 (v/v) humus. The chemical properties of the humus-mixed soil were as follows: pH 6.5, EC 385.6 ($\pm$14.5) µs cm$^{-1}$, organic matter 16.90 ($\pm$1.82) g kg$^{-1}$, total nitrogen (TN) 1.13 ($\pm$0.06) g kg$^{-1}$, total phosphorus (TP) 0.27 ($\pm$0.03) g kg$^{-1}$, total potassium (TK) 1.08 ($\pm$0.05) g kg$^{-1}$, available nitrogen (AN) 291.50 ($\pm$41.9) mg kg$^{-1}$, available phosphorus (AP) 3.66 ($\pm$0.22) mg kg$^{-1}$, available potassium (AK) 32.76 ($\pm$2.41) mg kg$^{-1}$.

The pH of the humus-mixed soil was adjusted according to *Soti et al. (2015)*. Briefly, a soil neutralization curve was generated to determine the amount of hydrated limes powder or ferrous sulfate to be added to the potting soil. Their levels required to increase or decrease soil pH to the desired levels were determined based on the regression equation resulting from pH measurement of the incubated soils. For pH 7.2, 9.0: $Y_{(pH\ value)} = 34.25X_{(glime/gsoil)} + 7.10$. For pH 5.5: $Y_{(pH\ value)} = -17.51X_{(g\ ferrous\ sulfate/gsoil)} + 6.26$. The pH-adjusted soils were irrigated and incubated in a greenhouse located at Kunming University (altitude 1,890 m; 24°58′N; 102°48′E). The greenhouse had a natural light with an average 25.4 ($\pm$12.1)°C temperature and 75 ~95% relative humidity during the experiment. Soil moisture content in the pots was maintained at 20% every 2 days. After 2 months, the soil pH values were determined.

## Seed collection, seedling

Seeds were collected from *A. adenophora* in an evergreen mixed forest located at Mao-Mao Qing, Xishan of Kunming, Yunnan Province, China (24°58′33.1″N102°37′04.9″E). The sampling area has been dominated by *A. adenophora* for the last 30 years (*Sun, Gao & Guo, 2013*) and has an average altitude of 2,200 m, a mean annual precipitation of 932.7 mm and a mean annual temperature of 15.6 °C. The seeds were collected from more than 10 individuals that were at least 5 m apart from one another, and stored at 5 °C after air-drying at room temperature. They were germinated in seed beds with the humus-mixed soil in December 2021 in the same greenhouse at Kunming University.

## Pot experiment design

Four pot planting treatments were designed. The soil samples with their pH adjusted to pH 5.5, 6.5 (original soil), 7.2 and 9.0 were used in pot experiments. According to *Huang (2000)*, these soils corresponded to acidic (pH 4.5–5.5), weakly acidic (pH 5.6–6.5), neutral (pH 6.5–7.5), and alkaline (pH 8.5–9.5) soils. In each pH treatment, three healthy seedlings that were approximately 10 cm tall, similar-sized were transplanted at equal distance from each other in plastic pots (height 20 cm, diameter 19 cm) containing 8 kg soils. Each pot was watered to 2/9th of soil maximum holding capacity 24 h prior to transplantation, and then at 1/9th of their maximum water retention capacity every 48 h to the end of the experiments, which were carried out in the same greenhouse for 180 days (from December 2021 to May 2022). Six replicate pots were used for each pH treatment for determinations of plant growth indices at 10, 90 and 180 days.

## Chemical analysis of soil samples

Chemical analyses of soil samples were carried out according to the protocols described by *Kong et al. (2017)*. Briefly, TN (total nitrogen), TP (total phosphorus), and TK (total potassium) were determined using the Kjeldahl method, the molybdenum blue colorimetric method, and the flame photometric method (*Kuo, 1996*), respectively. AN (available nitrogen), AP (available phosphorus), and AK (available potassium) were determined with the alkaline hydrolysis diffusion method, the molybdenum blue colorimetric method, and the flame photometric method (*Helmke & Sparks, 1996*) respectively. The soil pH (1:2.5 solution of soil to water) values were measured using a pH meter (Mettler-Toledo International Inc., Columbus, OH, USA). The soil EC (1:5 solution of soil to water) values were measured according 1:5 soil to water ratio conductivity method (*USDA, 1954*). Soil water holding capacity was determined according to the cutting ring method described by (*Chen et al., 2016*). All parameters were measured in triplicates (see "DNA extraction and PCR amplification of rhizosphere microbiomes" for more detail).

## Plant growth indices

Of the 18 plants of *A. adenophora* in the six pots of each pH treatment, 16 plants (excluding the highest and shortest ones) were chosen to determine above- and under-ground fresh and dry weights, plant heights and root lengths at 10, 90 and 180 days after seedling transplanting. These data were analysed for each plant. In order to determine their dry weight, plant components were placed in a hot air oven at 60 °C until a steady value was reached.

## Antioxidant enzyme activities and redox marker levels in the leaves of *A. adenophora*

Aliquots of 0.2 g leaf samples were homogenized in 1.8 mL of 0.1 M phosphate buffer (pH 7.0) on ice, followed by centrifugation at 10,000 rpm for 10 min. Triplicated leaf samples from different pots were used for activity analysis of each enzyme. The supernatants were collected, the activities of oxidative enzymes and levels of redox markers determined. These were superoxide dismutase (SOD), peroxidase (POD), catalase (CAT), glutathione (GSH) which controls reactive oxygen species and involves in detoxification of methylglyoxal, and lipid peroxidation marker malondialdehyde (MDA), and were analysed using the detection kits (Nanjing Jiangcheng Bioengineering Institute, Nanjing, China) A001-1-2, A084-3-1, A007-1-1, A061-2-1 and A003-1-2 respectively according to the manufacturer's instructions. Quantification of the activities of these enzymes were measured with a microplate reader (Thermo Fisher Scientific, Waltham, MA, USA).

## DNA extraction and PCR amplification of rhizosphere microbiomes

Rhizosphere soils of *A. adenophora* were collected by using the shaking root method (*Xia et al., 2023*). Equal amounts (80 g) of these from the three plants in a single pot were homogenized and divided into two aliquots. Rhizosphere soils were sampled at each soil pH taken at 10, 90 and 180 days after transplanting. One aliquot was air-dried and used for analysis of soil chemical properties. The other was quick-frozen in liquid nitrogen immediately and stored at −80 °C for microbial community analysis (only for soil samples

90 and 180 days after transplanting). DNA extraction, PCR amplification, and Illumina sequencing were carried out according to the protocols described by *Xia et al. (2021)*. Soil DNA was submitted to Majorbio Bio-Pharm Technology Co., Ltd., Shanghai of China for amplification and Illumina sequencing (NovaSeq PE 250) of the V3-V4 hypervariable region of the 16S rRNA genes and the ITS2 regions of fungal rRNA genes. The 16S rRNA and ITS amplicon sequences have been deposited in the NCBI Sequence Read Archive under the submission ID: SUB13856309 and BioProject ID: PRJNA1034221.

## Phylogenetic analyses of rhizosphere microbiomes

Phylogenetic analyses were carried out according to the methods described by *Xia et al. (2021)* with modifications. The V3–V4 amplicons of the 16S rRNA genes and ITS fragments were pair-end assembled and checked using Flash software (*Magoč & Salzberg, 2011*) to ensure that their sequences matched perfectly with the index sequences, had no more than one mismatch error present in the forward primer sequences, and trimmed sequences were longer than 200 bp. Then, QIIME (*Caporaso et al., 2010*) was used to analyse the 16S rRNA and ITS amplicons to generate ASV (amplicon sequence variant) clusters and perform alpha diversity analyses. The number of sequences per sample was normalized based on the number of sequences obtained from the smallest library for each community before analysis. The V3–V4 and ITS amplicon sequences were grouped into ASVs at the 97% identity threshold (3% dissimilarity levels) using the RDP classifier (Release 11.1; https://sourceforge.net/projects/rdp-classifier/) and Unite (Release 6.0; https://unite.ut.ee/index.php), respectively. Any ASV represented by ≤3 sequences was removed. Biodiversity indices, including the Chao1 index, Shannon index, and coverage ratios, were calculated with Mothur (*Schloss et al., 2009*) following the procedures provided and again applying a 97% identity threshold.

## Statistical analyses

The Kruskal–Wallis test was used to assess differences in soil chemical properties, leaf enzyme activities, plant growth indices, microbial community abundances, and diversities between different pH treatments. Pairwise-Wilcox test was used to determine the difference significance ($P < 0.05$). Pearson's correlation analysis was performed between microbial abundances and soil pH or growth time of *A. adenophora*, and between soil pH and soil nutrient concentrations. For Pearson's correlation analyses, the normality of data was confirmed by a Kolmogorov–Smirnov test. All analyses described above were performed with SPSS 17.0. The PCoA (principal coordinates analysis) based on the Bray–Curtis distance was chosen to perform the cluster analyses of the prokaryotic and fungal community composition between different pH treatments. db-RDA (distance-based redundancy) analyses base on Bray-Curtis distance of the correlations between soil pH, growth time and the composition of the rhizosphere prokaryotic and fungal communities of *A. adenophora* were performed with vegan in R packages. PERMANOVA analyses to differentiate the impact of soil pH and days of planting *A. adenophora* on the composition of the rhizosphere microbiomes of *A. adenophora* were carried out in R packages. Active microbial co-occurrence network analysis was conducted using the R packages. A Pearson's

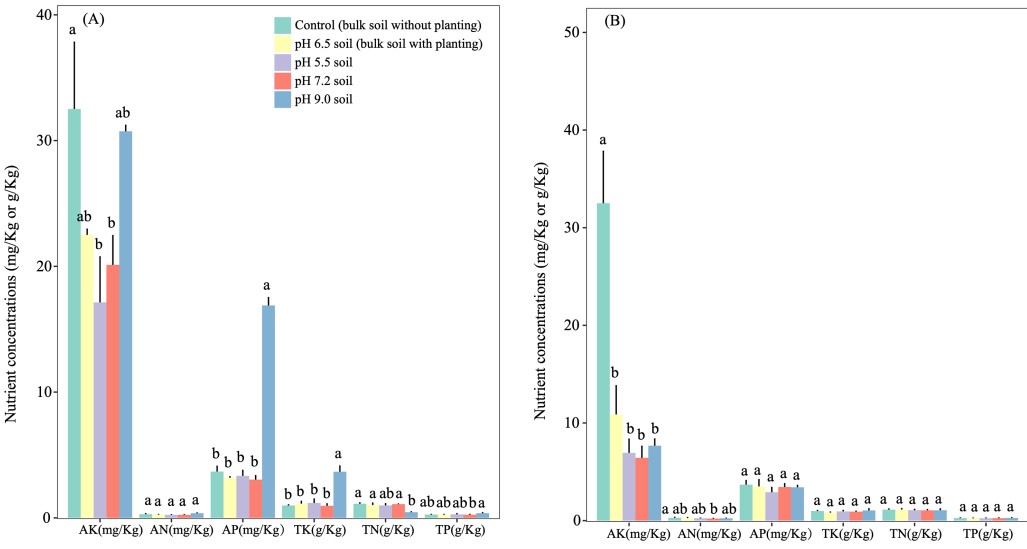

**Figure 1** **Effect of initial soil pH on chemical features of the rhizosphere soils of *Ageratina adenophora* after transplanting for 90 (A) and 180 (B) days.** Different lowercase letters within the same parameter indicates significance at the $P < 0.05$ level between different pH treatments.

coefficient of greater than 0.6 (or less than −0.6) and a significance level of less than 0.05 indicated a significant correlation. Topological features including number of edges, betweenness- and degree-centralization of each subnetwork were calculated for analysis of the distance-decay relationship of the prokaryotic and fungal co-occurrence patterns. Subsequently, network diagrams were generated using Gephi software (version 0.9.2) (*Bastian, Heymann & Jacomy, 2009*). Heatmaps showing changes in the degree-centrality values of the top 50 prokaryotic and fungal genera of the rhizosphere microbiomes of *A. adenophora* with soil pH and planting time (90 and 180 days) were modelled with pheatmap in R package.

# RESULTS

## Soil pH and its effects on the N, P, K concentrations of rhizosphere soil of *A. adenophora*

Soil pH affected the nutrient concentration in the rhizosphere soils. At day 90 (Fig. 1A), the rhizosphere soil of *A. adenophora* at pH 9.0 had higher ($P < 0.05$) available P and total K concentrations than those of the other three pH treatments, a lower ($P < 0.05$) total N content than those at pH 7.2 and 6.5, and a higher ($P < 0.05$) total phosphorus concentration than that at pH 7.2. At day 180 (Fig. 1B), no significant differences ($P > 0.05$) were seen between the total and available concentrations of N, P, and K among rhizosphere soils at the different pH values.

We also measured the pH values of the rhizosphere soils of *A. adenophora* after 90 and 180 days (Fig. 2). In all four soils they decreased after day 10, except in the pH 9.0 soil, which continued to decrease by day 90. Soil pH values in those with initial pH 5.5, 6.5 and

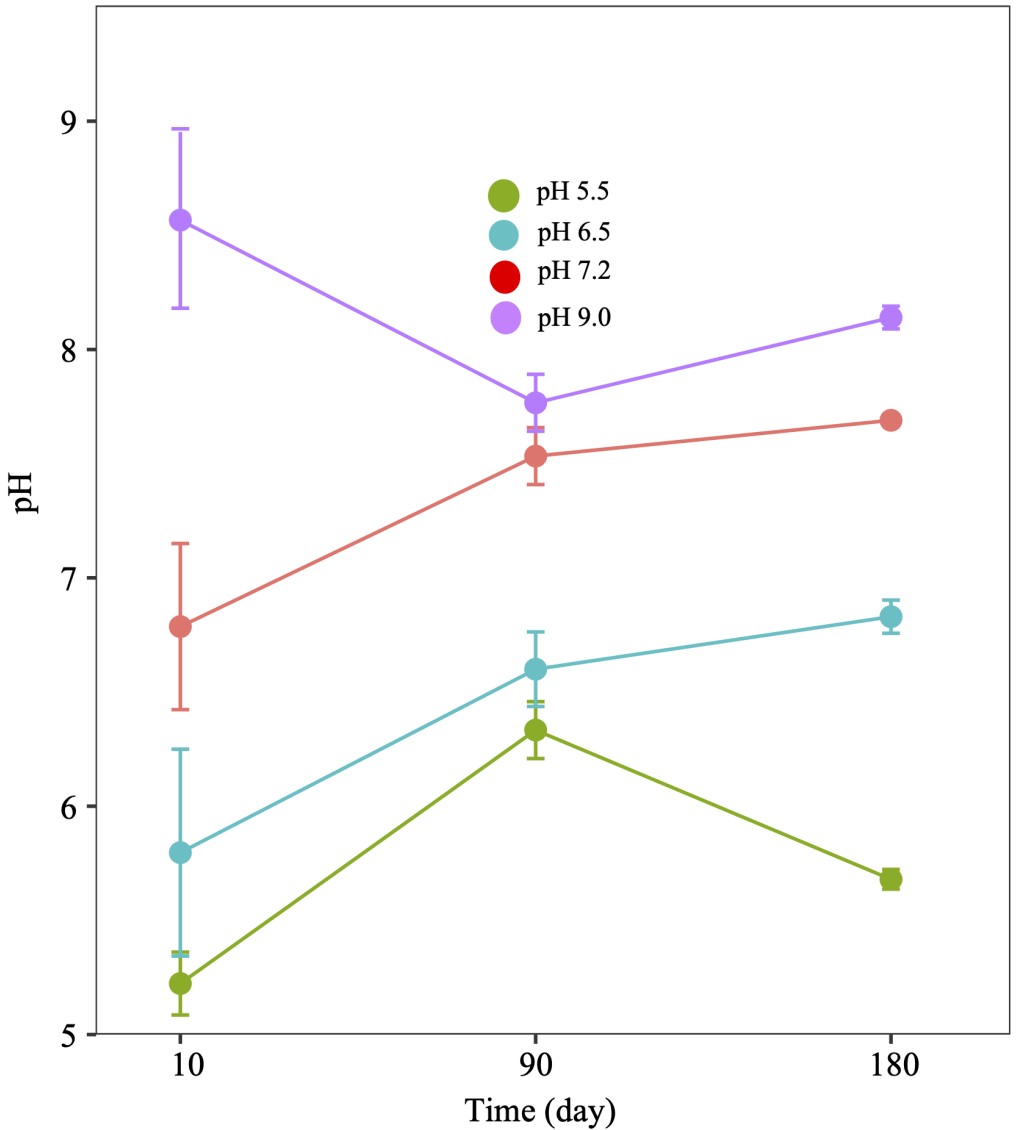

**Figure 2** Diagram of the pH change of the rhizosphere soils of *Ageratina adenophora* grown at different pH after transplanting for 10, 90 and 180 days.

7.2 had risen by day 90, while after day 180, soils with initial pH values of 6.5, 7.2, 9.0 had increased slightly, yet with soil at pH at 5.5, a decrease was recorded.

When a Spearman correlation between the rhizosphere soil pH values and the total and available concentrations of N, P, and K of the rhizosphere soils after days 0, 90 and 180 was performed, no significant ($P > 0.05$) correlations were apparent.

## Effects of soil pH on growth indices of *A. adenophora*

Soil pH also affected the growth indices of *A. adenophora* during the early and middle experimental periods, but not at its end. No significant differences in the growth indices of

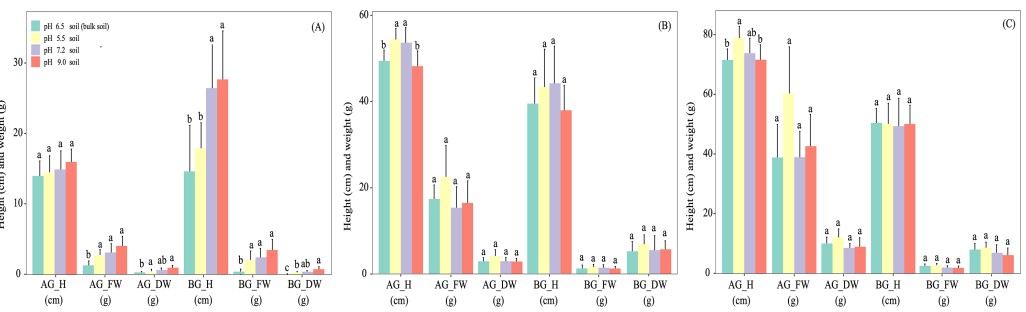

**Figure 3  Effects of soil pH on the growth indexes of *Ageratina adenophora* after transplanting for 10 (A), 90 (B) and 180 (C) days.** Different letters within the same parameter indicates significance at the *P* < 0.05 level between different pH treatments. AG_H, aboveground height; AG_FW, aboveground fresh weight; AG_DW, aboveground dry weight; BG_H, belowground height; BG_FW, belowground fresh weight; BG_DW, belowground dry weight.

*A. adenophora* were found in soils of different pH at day 180 (Fig. 3C), apart from a higher (*P* < 0.05) below-ground height in soil of pH 7.2 than in that in pH 5.5 soil at day 10 (Fig. 3A) and a higher (*P* < 0.05) above-ground height in pH 7.2 soils than those at pH 9.0 at day 90 (Fig. 3A).

## Effect of soil pH on enzyme activities in *A. adenophora* leaves at different soil pH

Soil pH had fluctuating effects on the activities of the *A. adenophora* leaf antioxidant enzymes. Thus, at day 90 (Fig. 4A), significant (*P* < 0.05) differences were detected in the activities of CAT (pH 9.0>5.5 >pH 7.2>6.5), GSH (pH 7.2 >pH 5.5) and SOD (pH 9.0 >pH 7.2). At day 180 (Fig. 4B), CAT activities in plants grown at pH 7.2 were greater (*P* < 0.05) than in those grown at pH 5.5.

## Effect of soil pH on the diversity of *A. adenophora* rhizosphere microbiome

Soil pH affected the diversity of prokaryotic (bacteria and archaea) communities in the rhizosphere soil of *A. adenophora* (Table 1). The Shannon indices of those grown in pH 5.5 and pH 9.0 soils were lower (*P* < 0.05) than those in soils at pH 7.2 after days 90 and 180 respectively. Soil pH did not affect (*P* > 0.05) the richness of the prokaryotic communities by days 90 and 180 (Table 1). However, except for that at pH 9.0, the Shannon indexes of those at soil pH 5.5, 6.5 and 7.2 after day 180 were significantly higher (*P* < 0.05) than those analyzed after day 90.

Soil pH also affected the diversity of rhizosphere fungal communities of *A. adenophora* (Table 2) but to a lesser extent. Thus, no statistically significant (*P* > 0.05) differences were apparent in the Shannon indices and richness of those at different soil pH at day 90. However, at day 180, both indices in soil pH 9.0 were lower (*P* < 0.05) than those in soil pH 7.2. while those detected at day 180 were not significantly (*P* > 0.05) different to those detected at day 90 (Table 2).

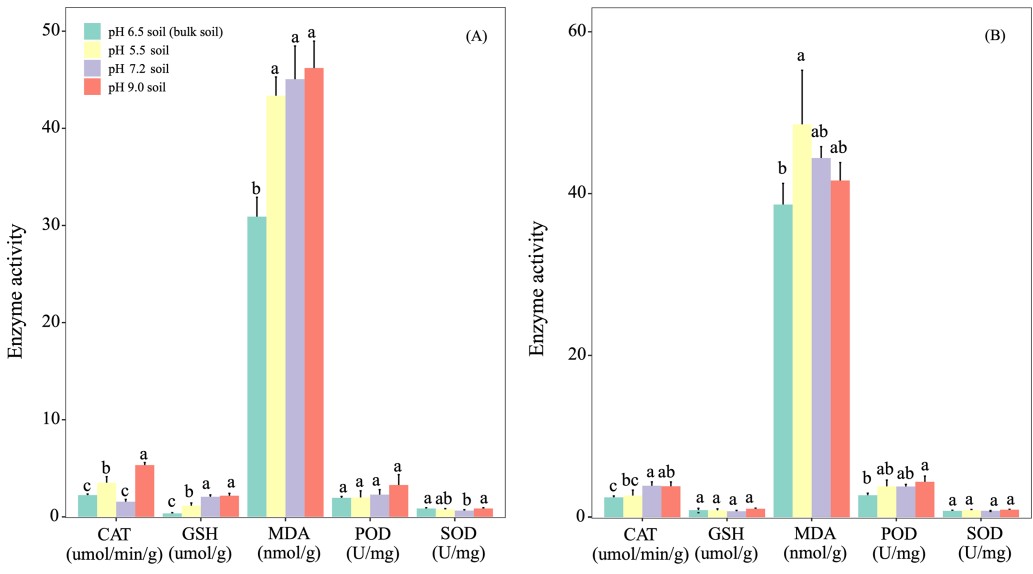

**Figure 4  Effects of soil pH on the antioxidant enzyme activities and levels of redox markers in the leaves of *Ageratina adenophora* after transplanting for 90 (A) and 180 (B) days.** Different letters within the same parameter indicates significance at the $P < 0.05$ level between different pH treatments. CAT, catalase; GSH, glutathione; MDA, malondialdehyde; POD, peroxidase; SOD, superoxide dismutase.

**Table 1  Diversity indices of the rhizosphere prokaryotic communities of *Ageratina adenophora* grown at different pH and for different times (90 and 180 days).** Different lowercase letters in the same column indicates a significant difference ($P < 0.05$) between the diversity indices of different treatments.

| Soil treatment group | Chao | Sobs | Shannon | Coverage |
|---|---|---|---|---|
| Bulk soil (control) | 2,995* ($\pm$615)$_b$ | 2,905 ($\pm$468)$_b$ | 6.71 ($\pm$0.09)$_c$ | 1 |
| pH 6.5+90 d planting | 2,872 ($\pm$356)$_b$ | 2.817 ($\pm$339)$_b$ | 6.64 ($\pm$0.13)$_{bc}$ | 1 |
| pH 5.5+90 d planting | 2,581 ($\pm$823)$_{ab}$ | 2,429 ($\pm$579)$_{ab}$ | 6.29 ($\pm$0.16)$_d$ | 1 |
| pH 7.2+90 d planting | 3,282 ($\pm$257)$_b$ | 3,066 ($\pm$200)$_b$ | 6.68 ($\pm$0.11)$_a$ | 0.99 |
| pH 9.0+90 d planting | 3,162 ($\pm$771)$_{ab}$ | 2,863 ($\pm$557)$_{ab}$ | 6.09 ($\pm$0.30)$_{abcd}$ | 0.99 |
| pH 6.5+180 d planting | 4,339 ($\pm$398)$_a$ | 3,969 ($\pm$266)$_a$ | 7.21 ($\pm$0.02)$_a$ | 0.99 |
| pH 5.5+180 d planting | 3,578 ($\pm$669)$_{ab}$ | 3,326 ($\pm$476)$_{ab}$ | 6.83 ($\pm$0.15)$_{bc}$ | 0.99 |
| pH 7.2+180 d planting | 4,012 ($\pm$362)$_a$ | 3,709 ($\pm$218)$_a$ | 7.05 ($\pm$0.06)$_{bd}$ | 0.99 |
| pH 9.0+180 d planting | 3,472 ($\pm$739)$_{ab}$ | 3,165 ($\pm$506)$_{ab}$ | 6.35 ($\pm$0.19)$_c$ | 0.99 |

The coverage values of both the prokaryotic (Table 1) and fungal rhizosphere communities (Table 2) in the 27 soil samples detected at days 90 and 180 were all greater than 0.99, indicating that the sequencing depth applied covered the diversity of both rhizosphere microbiomes. Each soil sample rhizosphere consisted of 3,041–5,370 bacterial and archaeal ASVs (Fig. 5A) and 320–818 fungal ASVs (Fig. 5B), with 141 identical prokaryotic and 75 fungal ASVs shared between them. We performed PCoA of the ASVs of all rhizosphere microbiomes using Bray-Curtis distance (Fig. 5C & Fig. 5D), which showed those of the three replicate soil samples in each soil pH were tightly clustered after days 90

**Table 2 Diversity indices of the rhizosphere fungal communities of *Ageratina adenophora* grown at different pH and for different times (90 and 180 days).** Different lowercase letters in the same column indicates a significant difference ($P < 0.05$) between the diversity indices of different treatments.

| Soil treatment group | Chao | Sobs | Shannon | Coverage |
|---|---|---|---|---|
| Bulk soil (control) | 496 ($\pm$44)$_d$ | 496 ($\pm$44)$_c$ | 3.1 ($\pm$0.16)$_c$ | 1 |
| pH 6.5+90 d planting | 640 ($\pm$80)$_{abcd}$ | 640 ($\pm$80)$_{abc}$ | 3.7 ($\pm$0.51)$_{abc}$ | 1 |
| pH 5.5+90 d planting | 796 ($\pm$34)$_b$ | 796 ($\pm$34)$_b$ | 4.3 ($\pm$0.07)$_b$ | 1 |
| pH 7.2+90 d planting | 685 ($\pm$95)$_{bd}$ | 685 ($\pm$95)$_{bc}$ | 4.1 ($\pm$0.51)$_{abc}$ | 1 |
| pH 9.0+90 d planting | 669 ($\pm$103)$_{abcd}$ | 669 ($\pm$103)$_{abc}$ | 4.1 ($\pm$0.28)$_{ab}$ | 1 |
| pH 6.5+180 d planting | 665 ($\pm$18)$_{bc}$ | 665 ($\pm$18)$_b$ | 4.3 ($\pm$0.20)$_b$ | 1 |
| pH 5.5+180 d planting | 1,031 ($\pm$56)$_a$ | 1,031 ($\pm$56)$_a$ | 4.7 ($\pm$0.13)$_a$ | 1 |
| pH 7.2+180 d planting | 916 ($\pm$15)$_a$ | 916 ($\pm$15)$_a$ | 4.9 ($\pm$0.05)$_a$ | 1 |
| pH 9.0+180 d planting | 677 ($\pm$88)$_{be}$ | 677 ($\pm$88)$_{bc}$ | 4.5 ($\pm$0.16)$_b$ | 1 |

and 180, with those of the former showing a clear separation along axis 1 and axis 2 from those at day 180 (Fig. 5C & Fig. 5D).

## Effect of soil pH on the composition of *A. adenophora* rhizosphere microbiome

Soil pH affected markedly the composition of the rhizosphere microbiome of *A. adenophora*. The prokaryotic communities (Fig. 6A, Table S1) in soils at pH 5.5 contained fewer ($P < 0.05$) *Firmicutes* (0.89% *vs* 2.38% at day 90; 1.10% *vs* 3.38% at day 180), *Thermotogota* (0.48% *vs* 1.84% at day 90) and more ($P < 0.05$) *Verrucomicrobiota* (1.57% *vs* 0.45% at day 90) than those in pH 7.2 soils. Those at pH 9.0 had more ($P < 0.05$) *Proteobacteria* (57.61% *vs* 51.71% at day 180) and fewer ($P < 0.05$) Planctomycetes (0.96% *vs* 2.43% at day 180) than those at soil pH 7.2 (Fig. 6A, Table S1). The fungal communities (Fig. 6C, Table S2) of those in soils at pH 5.5 had fewer ($P < 0.05$) *Rozellomycota* (0.25% *vs* 0.85%) than those at pH 7.2, and those at pH 9.0 contained more ($P < 0.05$) *Chytridiomycota* (19.23% *vs* 1.03% at day 180) and *Mortierellomycota* (16.94% *vs* 2.29% at day 180) than those at pH 7.2 (Fig. 6B, Table S2).

We compared the top 50 abundant genera in all 27 soil samples and found significant differences in their abundance (SDA) at each soil pH (Fig. 6B, Table S3). There were one-to-five (average 3.3) and three-to-seven (average 4.3) SDA genera among the prokaryotic communities of the pH 5.5, 6.5, 7.2 soils at days 90 and 180, respectively, and 12-28 (average 18.7) and eight-to-24 (average 14.7) SDA genera between these three soil pH soils and the pH 9.0 soil at days 90 and 180 respectively. The highest differences (28 and 24 SDA genera at days 90 and 180 respectively) were between the pH 5.5 and pH 9.0 soils.

Similarly, of the top abundant 50 fungal genera in all soil samples (Fig. 6D, Table S4), there were one-to-five (average 2.7) and two-to-12 (average 6.3) SDA genera among fungal communities in soils at pH 5.5, 6.5, 7.2 at days 90 and 180 respectively, and seven-to-19 (average 13.7) and three-to-nine (average 5.3) SDA genera between three soil pH and the pH 9.0 soil at days 90 and 180, respectively. The most difference, not surprisingly (19 and nine SDA genera at days 90 and 180 respectively) was seen between the pH 5.5 and the pH 9.0 soils.

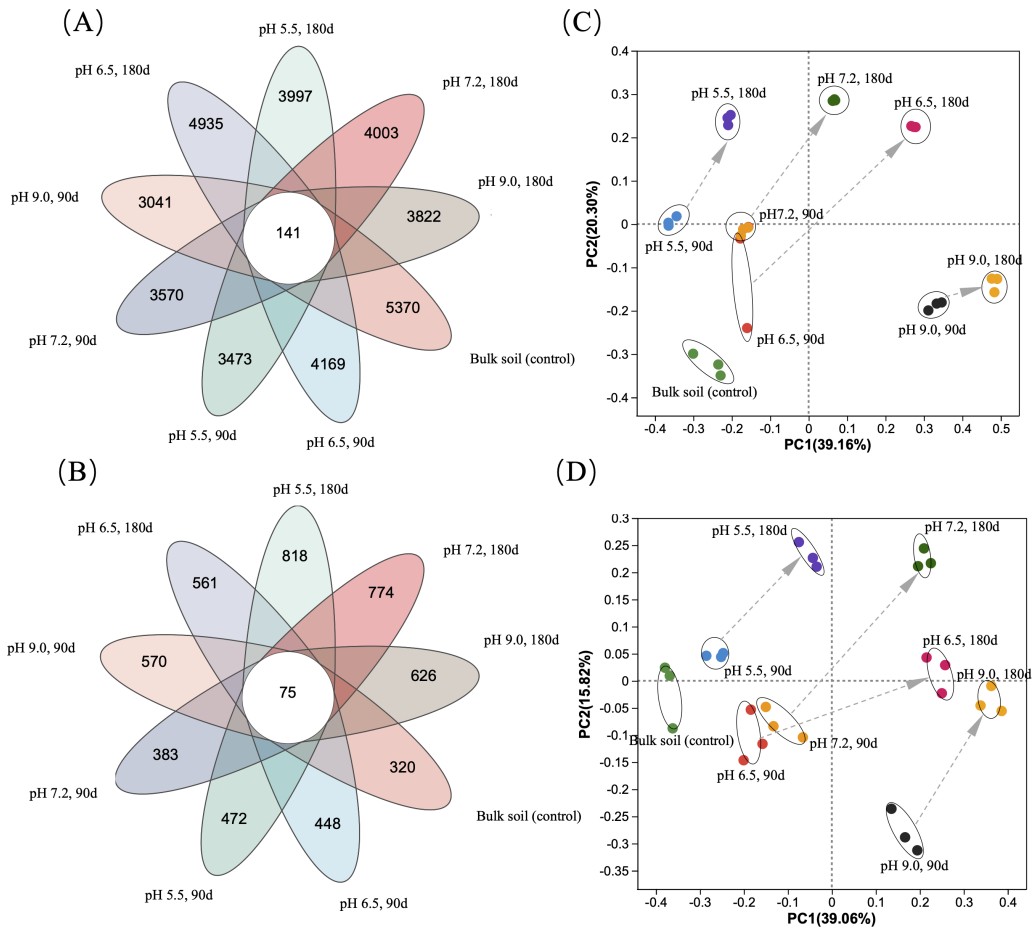

**Figure 5  Effects of soil pH on the distribution of prokaryotic and fungal ASVs and communities of the rhizosphere microbiome of *Ageratina adenophora* after transplanting for 90 and 180 days.** (A, C) Venn diagrams showing the occurrence of prokaryotic (A) and fungal (C) ASVs identified with 16S rRNA (V3-V4 region) and ITS fragment sequencing respectively of the rhizosphere microbiomes of *A. adenophora*. (B, D) Grouping of prokaryotic (B) and fungal (D) communities based on the principle component analyses of 16S rRNA (V3–V4 region) and ITS fragment sequences of the rhizosphere microbiomes of *A. adenophora*.

## Effects of soil pH on common pattern of the rhizosphere microbiome of *A. adenophora*

To further investigate the effects of soil pH on the composition and population interaction of these rhizosphere microbiomes, we analyzed correlations among the top 50 abundant prokaryotic and fungal genera in the three replicate soil samples at each pH treatment, and calculated values for their node-level topological features including total degree (edge) number, degree-, closeness- and betweenness-centrality. A correlation network diagram was constructed for both the abundant prokaryotic (Fig. 7A) and fungal (Fig. 7B) genera identified in all 27 soils samples. We found that soil pH affected the edge number and centrality values in both the prokaryotic (Fig. 8A) and fungal (Fig. 8B) networks. At day 90, the total edge numbers of the soil pH 5.5, 6.5, 9.0 in the prokaryotic networks and

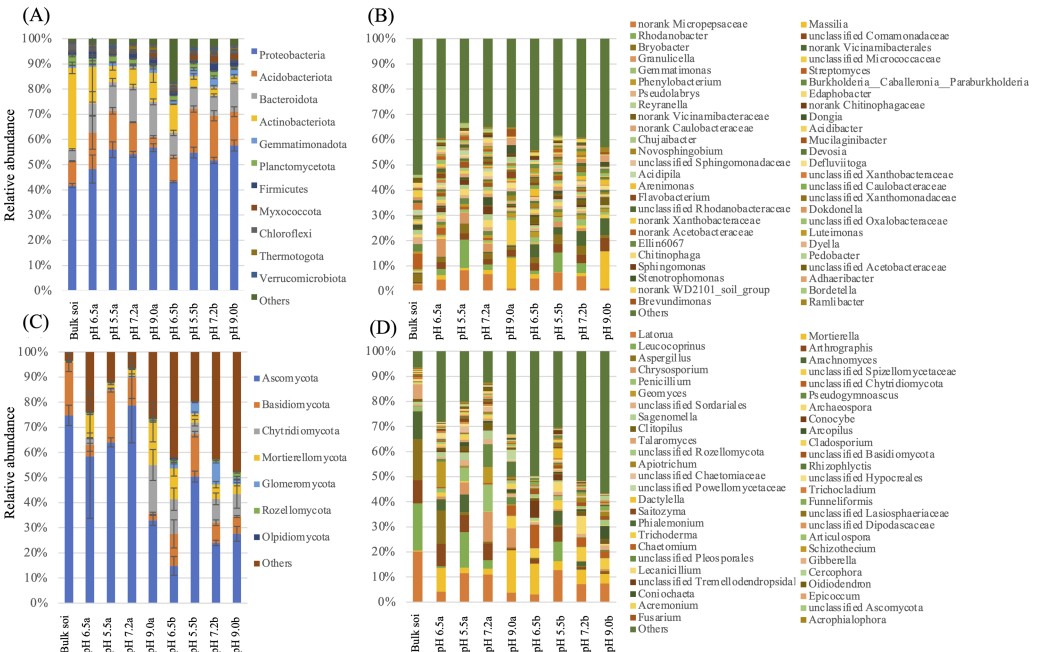

**Figure 6 Effects of soil pH on the composition of the rhizosphere microbiome of *Ageratina adenophora* after transplanting for 90 and 180 days.** (A, B) Phylum (A) and genus (B) composition of the prokaryotic communities characterized based on 16S rRNA amplicons (V3–V4 region) sequencing. (C, D) Phylum (A) and genus (B) composition of the fungal communities characterized based on ITS fragment sequencing.

the pH 6.5, 7.2, 9.0 soils in the fungal networks were all greater than that of the original bulk soil (Fig. 8). At day 180, total edge numbers for all other soil pH samples returned to the level of the control, except for the slightly higher total edge number in the pH 9.0 soil sample of the prokaryotic and the lower total edge number in the pH 7.2 soil sample of the fungal networks. Substantial differences were seen in the degree-centrality value of the top 50 dominant genera in rhizosphere soils among different pH treatments (Fig. S1). For example, the degree-centrality value of the bacterial genus *Devosia* in the bulk soil, the pH 6.5, 5.5, 7.2, 9.0 soils at days 90 and 180 were 0.37, 0.43, 0.21, 0.29, 0.63, 0.35, 0.27, 0.29, 0.49, respectively (Fig. S1A).

## Correlation effects between soil pH and *A. adenophora*'s growth time on the composition of the rhizosphere microbiome

We analyzed possible correlations between soil pH and *A. adenophora* growth time (90 and 180 days) on the ASV compositions of the prokaryotic (Fig. 9A) and fungal (Fig. 9B) communities. Analyses showed that both significantly ($P = 0.001$) affected the composition of the prokaryotic (Fig. 9A) and the fungal (Fig. 9B) communities. Moreover, the angles between the pH and planting-time vectors for the prokaryotic (Fig. 9A) and fungal communities (Fig. 9B) were 65.9 and 70.1 degrees, respectively, indicating that their impacts were only weakly positively correlated. Furthermore, we differentiated the impact of pH and days of planting *A. adenophora* on the composition of the rhizosphere microbiomes of
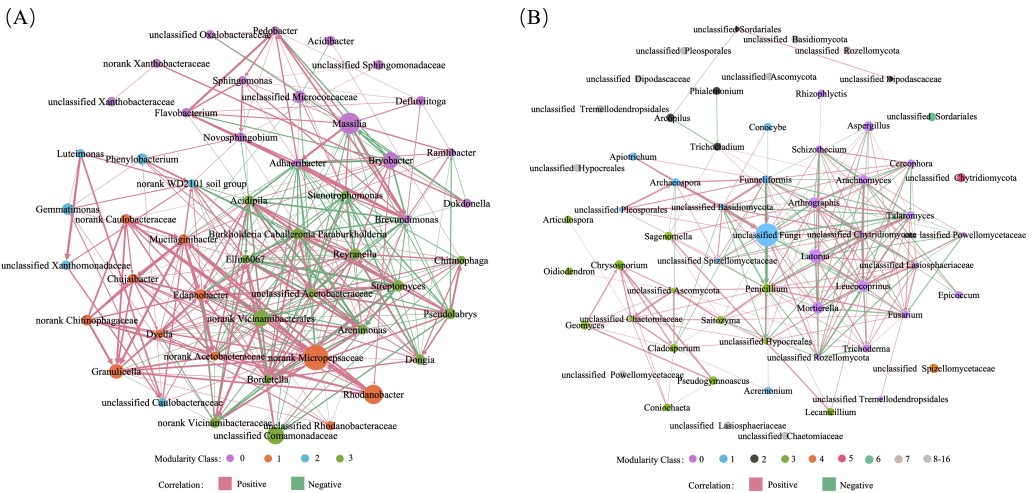

**Figure 7** Co-occurrence networks of the active microbial communities in the rhizosphere soils of *Ageratina adenophora* grown at different pH for 90 and 180 days. (A) Prokaryotic network. (B) Fungal network. Each co-occurrence network was based on the top 50 abundant genera in all 27 soil samples with a Pearson's coefficient > |0.6| and a *P* value <0.05 between ASVs. The pink lines represent significant positive correlations, and the green lines represent significant negative correlations. A node represented a genus, and the node size was proportional to the number of connections.

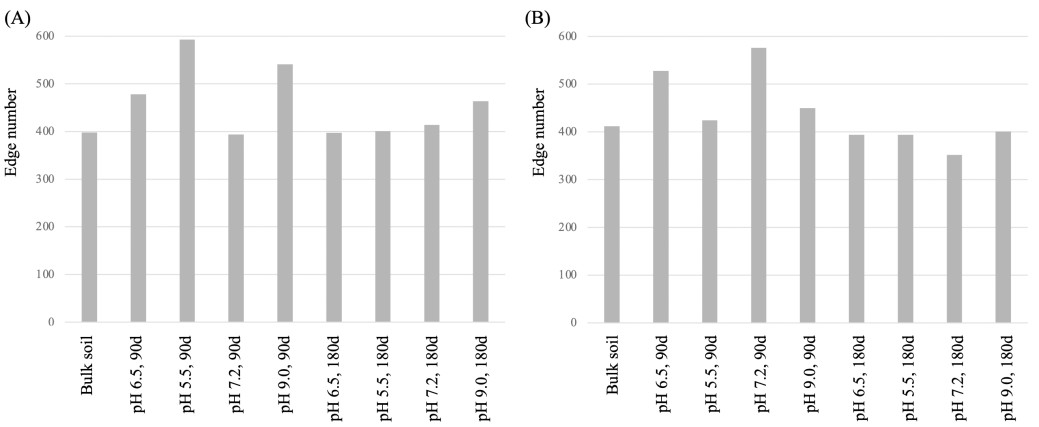

**Figure 8** Diagrams of total edge numbers in the co-occurrence networks of the active microbial communities in the rhizosphere soils of *Ageratina adenophora* grown under different soil pH conditions for 90 and 180 days. (A) Prokaryotic network. (B) Fungal network. The total edge numbers were calculated in co-occurrence network analyses based on the top 50 abundant genera in the triplicate soil samples of a pH treatment with a Pearson's coefficient > |0.6| and a *P* value < 0.05 between ASVs.

*A. adenophora* by using PERMANOVA analyses. The results showed that soil pH explained 62.2% and 52.8% effects in the prokaryotic and the fungal communities respectively, and the days of planting *A. adenophora* explained 37.8% and 47.2% effects in the prokaryotic and the fungal communities respectively.

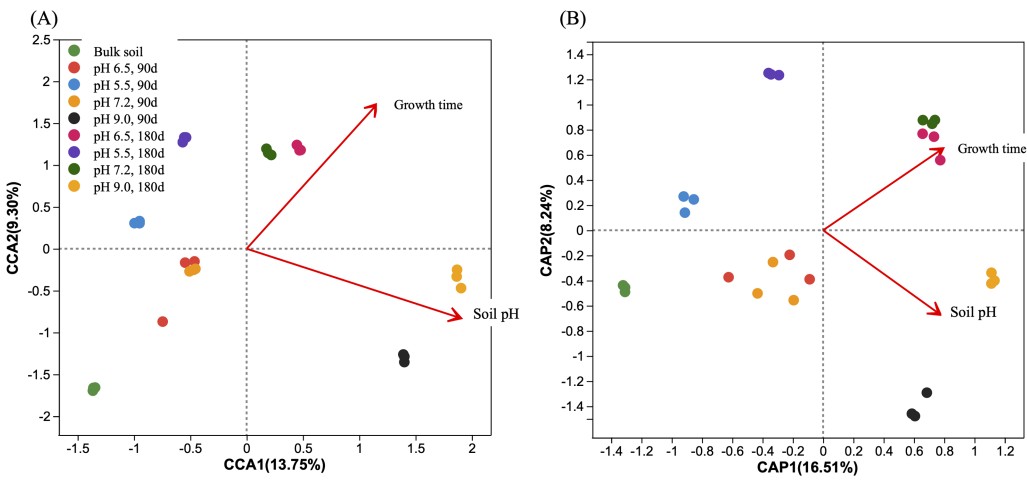

**Figure 9** Db-RDA analyses based on Bray-Curtis distance of the correlations between soil pH, planting time of *Ageratina adenophora* and composition of the rhizosphere microbiomes of *A. adenophora* grown at different pH for 90 and 180 days. (A) Prokaryotic communities. (B) Fungal communities. The cycles filled in with different colors represent the rhizosphere microbial communities of *A. adenophora* with different pH treatments. The red arrow represents soil pH and planting times, and the length of the arrow is proportional to their influential degree. The angel between soil pH and planting times represents their correlation relationship (sharp angle, positive correlation; blunt angle, negative correlation; right angel, no correlation). The distance of the projection point from the origin represents the relative influence of the soil pH and planting time on the distribution of the microbial communities.

## DISCUSSION

### Effects of soil pH on growth of *A. adenophora*

The soil pH is known to affect its biological, chemical, and physical properties, which determines the availability of nutrients. Therefore, soil pH is recognized as the most important soil factor for plant growth (*Neina, 2019*). Studies on the effects of soil pH on plant growth have focused mainly on crops, while limited information has been obtained on how it affects the growth of invasive alien plants. We studied the effects of acidic (pH 5.5), weakly acidic (pH 6.5), neutral (pH 7.2) and alkaline (pH 9.0) soils on growth of the weed *A. adenophora* by artificially changing the soil pH. Our results showed that significant differences in growth indices existed in the early (below-ground height of *A. adenophora* between pH 7.2 and pH 5.5 soils at day 10) or middle stages (above-ground height of *A. adenophora* between pH 7.2 and pH 9.0 soils at day 90) of the experiment. However, after 180 days, there were no statistically significant differences in above- and underground fresh and dry weights, plant heights, and root lengths of *A. adenophora* between those growing in the acid and alkaline soils and those in the neutral pH soil (Fig. 3). These data indicate that *A. adenophora* has a wide pH tolerance range and is able to grow normally in both acidic and alkaline soils. As discussed previously, the ideal soil pH for plant growth is between 6.5 and 7.5 where it is easier for plants to obtain most of the necessary soil nutrients. Thus, *Soti et al. (2015)* used similar pot experiments to study the effects of soil pH (pH 4.5 to 8.0) on the invasive alien species *Lygodium microphyllum* in Florida. They found that after 60-days, plant biomasses, relative growth rates, photosynthesis, and special

leaf areas of *L. microphyllum* growing at pH 5.5 and 6.5 were significantly higher than those of the other soil pH values. Furthermore, *Gentili et al. (2018)* examined the effect of soil pH in pot experiments (pH 5, 6, and 7) on the growth of the European invasive species *Ambrosia artemisiifolia* (ragweed). Their data showed that plant heights growing at pH 5 and 6 were significantly lower than those growing at pH 7. Thus, the unusual pH tolerance of *A. adenophora* reported here may be one crucial reason for its successful invasion, and suggests that attempts use soil pH to control its invasion by changing the soil pH (for example, using lime) will fail.

## Effects of soil pH on leaf enzyme activities and redox marker levels, and nutrient concentrations of rhizosphere soil

In the pot experiments we also measured enzyme activity and redox marker levels in the leaves of *A. adenophora*. When we compared the activities of four antioxidant enzymes and levels of two redox markers in the leaves of *A. adenophora* grown in acid and alkaline soils and those in neutral pH soils for 90 and 180 days, significant differences existed in the activities of CAT and the levels of GSH (Fig. 4). However, at day 180 only CAT activities in those grown at pH 5.5 were lower ($P < 0.05$) than those grown at pH 7.2. We also monitored the change of nutrient concentrations in the rhizosphere soils of *A. adenophora* grown at different soil pH. Significant differences were found in AP and TK, TP and TN between the rhizosphere soils of *A. adenophora* grown at pH 9.0 and 7.2 at day 90. No differences ($P > 0.05$) were seen at day 180.

## Diversity, composition and interaction of the rhizosphere microbiota of *A. adenophora* grown at different soil pH

The rhizosphere is where plants absorb soil nutrients and its microbiome has an important role in soil nutrient cycling and availability to plants. Its structure and function are influenced by plant rhizosphere exudates (*Sasse, Martinoia & Northen, 2018*). Estimates suggest that plants secrete 20% of the carbon and 15% of nitrogen they fix into the rhizosphere (*Haichar Fel et al., 2016*; *Venturi & Keel, 2016*), thus providing energy and nitrogen sources for microbial growth there. These include simple molecules, like sugars and organic acids, as well as plant secondary metabolites and complex polymer secretions such as mucilage. The composition and quantity rhizosphere exudates are known to vary with plant species, its developmental stage, and external abiotic factors (*Sasse, Martinoia & Northen, 2018*). *A. adenophora* can change both the composition and structure of the soil microbial community through rhizosphere exudates and litter degradation during its invasion process (*Liu et al., 2010*).

In this study, we analyzed the diversity and composition of rhizosphere microbiomes of *A. adenophora* grown at different soil pH for 0, 90 and 180 days by using Illumina high throughput sequencing. We showed that both rhizosphere microbiome diversity (Tables 1 and 2) and composition (Figs. 5 and 6, Tables S1–S4) under different soil pH conditions changed over 180 days. Under acid and alkaline conditions, the rhizosphere microbiomes differed in their diversity, phylum and generic compositions, and population interactions to relieve pH stress. We also revealed that soil pH had a greater impact on the diversity and composition of the prokaryotic rhizosphere communities than those of the fungal

communities (Tables 1 and 2, Tables S1–S4). This observation is generally in line with reports by *Rousk et al. (2010)* who found in their long-term liming experiment that both the relative abundances and diversities of bacteria were positively related to soil pH (pH 4.0−8.3), while relative abundances of fungi were unaffected by pH and fungal diversity was only weakly related to soil pH. This may be due to the resistance of fungi to an acidic pH *versus* bacteria and archaea (*Guo et al., 2022*).

Furthermore, the importance of each prokaryotic or fungal genus in the rhizosphere networks of *A. adenophora* varied under different soil pH values and at different periods (90 and 180 days) at the same pH (Fig. S1) based on the degree-centrality values of key microbial genera. This effect is proportional to their importance in formation of microbial networks (*Ma et al., 2016*). The over-time effects of soil pH on the diversity and composition of the rhizosphere microbiomes of *A. adenophora* is shown in the PCoA diagrams, where the prokaryotic (Fig. 5C) and fungal (Fig. 5D) communities at day 180 are clearly separated from those at day 90 for each soil pH.

Our microbial network analyses also showed that *A. adenophora* growing in the acid and alkaline soils differentially adjusted the interactions between its dominant genera in the rhizosphere prokaryotic and fungal networks (Fig. 8). The edge number between two nodes (genera) represents the network density, which is proportional to the magnitude of the interaction between two nodes (genera), and the total edge number indicates the stability of the microbiome. Compared with the bulk soil, interactions among the abundant prokaryotic (Fig. 8A) and fungal (Fig. 8B) genera in the rhizosphere microbiomes of *A. adenophora* growing in soils at all four soil pH values for 90 days were all enhanced, which suggests that the rhizosphere microbes displayed higher levels of interactions than did bulk soil communities (*De Angelis et al., 2009*). However, in the prokaryotic networks, more interactions appeared to be enhanced under acid and alkaline conditions, with total edge numbers being 1.51 and 1.37 times higher than under the neutral pH condition, respectively (Fig. 8A). In contrast, in the fungal networks, the interactions under acid and alkaline conditions were less, with total edge numbers of 0.74 and 0.78 times of that under neutral pH conditions, respectively (Fig. 8B). These results suggest that the stability of the prokaryotic networks in the rhizosphere microbiomes of *A. adenophora* growing in both acid and alkaline soils were enhanced, while those of the fungal networks were reduced. Interestingly, interaction levels between the abundant genera in both the prokaryotic and fungal networks at different soil pH mostly returned to the level of the bulk soil after 180 days, indicating that the adjustment of soil pH to the interactions of microbial networks of *A. adenophora*'s rhizosphere microbiomes was complete.

Changes in both structure and diversity of the rhizosphere microbiome at different soil pH were seen with (*Sasse, Martinoia & Northen, 2018*) *A. adenophora*. Our correlation analyses show that both soil pH and *A. adenophora*'s growth time significantly ($P = 0.001$) affected the compositions of the prokaryotic (Fig. 9A) and fungal communities (Fig. 9B) in the rhizosphere of *A. adenophora*. However, soil pH and planting time were weakly correlated. And soil pH had more impact than *A. adenophora*'s growth time on the composition of the prokaryotic and fungal communities. Acidic (pH 5.5) and alkaline (pH 9.0) soils impacted the early- and middle-stage growth of *A. adenophora*. However,

under these pH conditions *A. adenophora* changed the composition and structure of its rhizosphere microbiome through its rhizosphere exudates, improved soil nutrient supply thus eliminated the negative impact of soil pH, and maintained a normal growth at the end of the pot experiment.

## CONCLUSIONS

We show here that *A. adenophora* has a strong pH tolerance being able to grow normally in both acidic (pH 5.5) and alkaline (pH 9.0) soils. When growing in acidic and alkaline soils, *A. adenophora* altered the composition, diversity, and interactions of its rhizosphere microbiome especially the prokaryotic community. These changes helped to maintain a balanced nutrient supply to *A. adenophora*, allowing it to successfully adapt to pH stress in acid and alkaline soils. Thus, the unusual pH tolerance of *A. adenophora* may be one crucial reason for its successful invasion ability. These results suggest that attempts use soil pH to control its invasion by changing the soil pH will fail.

### Funding
This work was supported by the National Science Foundation of China (NSFC) grant (31860029 and 31760178). The funders had no role in study design, data collection and analysis, decision to publish, or preparation of the manuscript.

### Grant Disclosures
The following grant information was disclosed by the authors:
National Science Foundation of China (NSFC): 31860029, 31760178.

### Competing Interests
There are no conflicts of interest to report with regard to the information described in the manuscript.

### Author Contributions

- Yun Xia conceived and designed the experiments, analyzed the data, prepared figures and/or tables, authored or reviewed drafts of the article, and approved the final draft.
- Junna Feng analyzed the data, prepared figures and/or tables, and approved the final draft.
- Hongbo Zhang performed the experiments, analyzed the data, prepared figures and/or tables, and approved the final draft.
- Deyu Xiong performed the experiments, prepared figures and/or tables, and approved the final draft.
- Lingdong Kong performed the experiments, prepared figures and/or tables, and approved the final draft.
- Robert Seviour conceived and designed the experiments, authored or reviewed drafts of the article, and approved the final draft.

- Yunhong Kong conceived and designed the experiments, analyzed the data, prepared figures and/or tables, authored or reviewed drafts of the article, and approved the final draft.

### DNA Deposition

The following information was supplied regarding the deposition of DNA sequences:

The 16S rRNA and ITS amplicon sequences are available in the NCBI Sequence Read Archive: PRJNA1034221.

### Data Availability

The raw data of plant growth, leaf enzyme activity and soil characteristics are available in the Supplementary File.

### Supplemental Information

Supplemental information for this article can be found online at http://dx.doi.org/10.7717/peerj.17231#supplemental-information.

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
