# Peer review of "Effects of soil pH on the growth, soil nutrient composition, and rhizosphere microbiome of Ageratina adenophora"

_PeerJ, doi:10.7717/peerj.17231_

## Round 0.1 · original submission · Major Revisions

Dear Dr. Kong

Thank you for your submission to PeerJ.

Subsequent to going through the review reports, I am of the opinion that your article - Effects of soil pH on the growth, soil nutrients and rhizosphere microbiome of weed Ageratina adenophora -requires a number of Major Revisions.

Accordingly, you are advised to thoroughly revise the manuscript, keeping all comments and suggestions in mind. You should particularly be careful while revising the Materials and Methods section of the article so that the reviewers concerns are appropriately addressed. Furthermore, try to maintain coherence among different sections of the article in order to improve the quality. Some further improvements in Discussion section are also needed.

It is pertinent to mention that your revised manuscript will be evaluated again in order to ensure that you have made necessary revisions, considering the comments and suggestions.

Hope to receive the revised manuscript in due course.

With regards

**Language Note:** The review process has identified that the English language must be improved. PeerJ can provide language editing services - please contact us at [email protected] for pricing (be sure to provide your manuscript number and title). Alternatively, you should make your own arrangements to improve the language quality and provide details in your response letter. – PeerJ Staff

Reviewer 1 ·

Basic reporting

Overall, the study has some interesting findings but needs major and thorough revision.
Specific comments are mentioned in the Word file.

Experimental design

It's well planned but needs clear information as mentioned in the Word file

Validity of the findings

The study is not repeated

Additional comments

There are many comments mentioned in the word file and in the corrected PDF

Annotated reviews are not available for download in order to protect the identity of reviewers who chose to remain anonymous.

Reviewer 2 ·

Basic reporting

• English language must be improved and the introduction lacks connection between the statements.
• Abstarct lines 32-34 is not clear.
• Abstract must included with quantifiable results. Generic statements may be avoided.
• Give some background information on soil conditions favourable for A. adenophora and also what is the soil conditions in the weed infected areas.
• Mention about mode of spreading and life span of spiecies.
• Highlight the economic loss due to the species. Why it is important to control or how soil microbiome influences the weed species.
• What were the hypothesis and research questions of the study? Mention them in the introduction section
• Mention about What is research questions that is going to be addressed from this study?
• The figures are relevant, but the clarity of the figures must be improved.
• Figure 1. Title what do you mean by physiochemical paramaters. You mentioned only chemical paramaters. Kindly rectify the figure title.
• Figure 5 and 6 is not readble at all. Very poor quality figures.
• Literature well referenced & relevant.

Experimental design

• Original primary research within the Scope of the journal.
• What is the EC of initial soil samples
• Give the regression equation used for maintaining the pH levels in the soil
• Mention how much lime and sulour is used to achieve desirable soil pH in the study.
• Line 140; physiochemical analysis is not correct as it is only chemical properties of soil.
• Line 149-153: clearly write about ho samples were drawn. How 16 plants were selected from 6 pots? It is confusing. Clearly mention how many plants in each pot and each pH treatment were collected for recording the observations.
• What statistical design is used for the ANOVA.
• Two way ANOVa could have been used to show case the impact of pH and DAP.

Validity of the findings

• Figure 1: mention Y axis Lables with units
• Line 227-228: It is mentioned that at 180DAP, there is no difference in any nutrients across all pH conditions. But in figure available ka and N is showing difference as per DMRT ranking.
• Lines 229-233: very confusing statements, write the results properly.
• How come immedialty after 10days of transplating the growth of plant is differed and there was no difference at 90and 180DAP.
• There is no reasons explained in discusiion for significant changes at 10DAP.
• Conclusion line 420-421: it is mentioned that changing pH didinot affect the enzyme activity but pH 9 has influenced the enzymes as shown in figure 4
• Lines 409-414 it mentioned no significant difference in enzymes at 90a dn 180DAS except CAT. But figure 4 shows sisgnificant difference of other enzymes too.
• What is key findings, conclusion is very generic. What pH is suitable for weed species, what soil pH makes unsuitable for growth of this species. Mention the conlusion of specifically.

Additional comments

Effects of soil pH on the growth, soil nutrients and rhizosphere microbiome of weed Ageratina adenophora is interesting and the results says it is tolerant to wide range of pH inclusing acidic to alkaline. The study lacks proper hypothesis and research questions. The introduction must be carefully checked and there is a lack of connection between statements. Mentodology is very superficial and needs detaled presentation especially experimentation part. Experimental design should be mentioned and two way ANOVa may be tried to differentiate the pH and DAP effect. In many instance, authours mentioned there is no difference but actualy diffrences exhists. Conclusion required to rewrite with significant findings. Figures quality must be improved.

Reviewer 3 ·

Basic reporting

The article is written in acceptable scientific language. The Introduction section sufficiently substantiates the relevance of the research conducted. However, the influence of soil pH on the activity of photosynthetic enzymes (line 99) should be excluded from the purposes of the article, because The article does not provide data on this parameter.

Experimental design

The methods presented in the article are adequate for the study. However, several shortcomings prevent the reproducibility of these methods from being ensured.
Thus, in the subsection Seed collection, seedling, and pot experiments, it remains unclear how the water-holding capacity of the soil was determined. In addition, in this subsection, the total number of plants and the number of replicates for each pH value remain unclear from the description of the method.
In the subsection Activities of photosynthetic and oxidative enzymes in the leaves of A. adenophora, the abbreviation POD given on line 158 does not refer to the enzyme polyphenol oxidase. This abbreviation is used for the enzyme peroxidase. The abbreviation PPO is used for polyphenol oxidase. Also, this subsection does not describe the determination of the activity of photosynthetic enzymes.
In the subsection DNA extraction and PCR amplification of rhizosphere microbiomes, the volume of rhizosphere soils taken for the experiment remains unclear from the description of the method.

Validity of the findings

The article does not provide results on the effect of soil pH on the photosynthetic activity of A. adenophora. There are also no results on changes in polyphenol oxidase activity. The conclusions presented in the article do not meet the objectives of the article.

---

## Round 0.2 · accepted · Accept

Dear Dr. Kong,

Thank you for your submission to PeerJ.

I am writing to inform you that your manuscript - Effects of soil pH on the growth, soil nutrient composition, and rhizosphere microbiome of Ageratina adenophora - has been Accepted for publication.

Congratulations!

Reviewer 3 ·

Basic reporting

The article is written in acceptable scientific language. The Introduction section sufficiently substantiates the relevance of the research conducted.

Experimental design

All the shortcomings in the description of the methods were eliminated.

Validity of the findings

All the shortcomings specified in the review were eliminated.